# Understanding the Knowledge, Attitudes, and Practices of Healthy Eating among Adolescents in Chongqing, China: An Empirical Study Utilizing Structural Equation Modeling

**DOI:** 10.3390/nu16010167

**Published:** 2024-01-04

**Authors:** Ke Jiang, Laixi Zhang, Changxiao Xie, Zhourong Li, Zumin Shi, Manoj Sharma, Yong Zhao

**Affiliations:** 1School of Public Health, Chongqing Medical University, Chongqing 400331, China; 2021110631@stu.cqmu.edu.cn (K.J.); 2021120819@stu.cqmu.edu.cn (L.Z.); 2021120781@stu.cqmu.edu.cn (Z.L.); 2Research Center for Medicine and Social Development, Chongqing Medical University, Chongqing 400016, China; 3Research Center for Public Health Security, Chongqing Medical University, Chongqing 400016, China; 4Department of Nutrition and Food Hygiene, West China School of Public Health, Sichuan University, Chengdu 610041, China; xie_cx2023@163.com; 5Human Nutrition Department, College of Health Sciences, QU Health, Qatar University, Doha P.O. Box 2713, Qatar; zumin@qu.edu.qa; 6Department of Social and Behavioral Health, School of Public Health, University of Nevada, Las Vegas, NV 89119, USA; manoj.sharma@unlv.edu; 7Department of Internal Medicine, Kirk Kerkorian School of Medicine, University of Nevada, Las Vegas, NV 89102, USA; 8Chongqing Key Laboratory of Child Nutrition and Heath, Children’s Hospital of Chongqing Medical University, Chongqing 401147, China

**Keywords:** healthy eating, knowledge, attitudes, practices, structural equation modeling, adolescents

## Abstract

Healthy eating is crucial for optimal growth, development, and the prevention of chronic diseases in adolescents. Our study aimed to develop a comprehensive structural equation model (SEM) to analyze the relationships between knowledge, attitudes, practices (KAP), and healthy eating among adolescents in Chongqing. An online questionnaire was administered to middle school students in 39 districts and counties of Chongqing, China from 2 December to 15 December 2021 to collect sociodemographic and KAP information. SEM models were constructed to examine the relationships between educational environment and healthy eating knowledge, attitudes, and practices. The Monte Carlo test was employed to assess the significance of the mediating effect of relevant variables. The study included 139,832 adolescents aged 14.8 ± 0.7 years, with a 14% rate of overweight and obesity. Health and nutrition knowledge averaged a score of 3.60 ± 1.50 (correct rate: 60%). Participants had positive attitudes (average score: 13.61 ± 2.29) and engaged in healthy eating practices (average score: 12.06 ± 3.08). The results of the SEM revealed a significant influence of the educational environment on adolescents’ knowledge (β = 0.235, *p* < 0.001) and attitude towards healthy eating (β = 0.143, *p* < 0.001). Knowledge exhibited positive effects on both attitude (β = 0.225, *p* < 0.001) and practice (β = 0.054, *p* < 0.001), while attitude exhibited positive effects on practice behavior (β = 0.565, *p* < 0.001). The indirect effect of knowledge on practices through attitude was more substantial than the direct effect (ratio 2.361). Our study highlights the mediating role of attitude between healthy eating knowledge and practices. A significant association exists between a favorable educational environment and improved knowledge and positive attitudes toward healthy eating among adolescents. In the future, nutrition and health education should prioritize exploring effective ways to translate knowledge into practices.

## 1. Introduction

Adolescence is a crucial stage characterized by rapid physical, cognitive, and socio-emotional development occurring between the ages of 10 and 19 [1]. During this phase, individuals establish habits and behaviors that can have a profound impact on their long-term health outcomes [2]. One such behavior of paramount importance is healthy eating.

Healthy eating is crucial for optimal growth, development, and the prevention of chronic diseases in adolescents [3]. It involves consuming a well-balanced diet that provides essential nutrients while limiting the intake of unhealthy foods. However, the global prevalence of unhealthy dietary behaviors among adolescents, including excessive consumption of processed foods, sugary beverages, and snacks, as well as inadequate intake of nutritious foods such as fruits and vegetables, remains a substantial public health concern [4,5]. Studies have indicated that the eating behaviors of the majority of adolescents fall short of healthy dietary recommendations [6,7]. This can potentially account for the increasing prevalence of adolescent obesity [8,9], which in turn elevates the risk of non-communicable diseases (NCDs) in adulthood [10,11]. Therefore, adolescence plays a crucial role in the formation of dietary habits, underscoring the significance of health education [12].

The Knowledge, Attitude, Belief, and Practices (KABP/KAP) theory is an extensively utilized and well-developed cognitive theory in the field of health education research [13]. It posits that the process of individual health behavior encompasses the acquisition of knowledge, the development of attitudes, and the formation of behaviors. Individuals will engage in healthy behaviors only when they possess a thorough understanding of health-related knowledge and cultivate positive and accurate beliefs and attitudes [14]. Based on this theory, adolescents’ knowledge of nutrition can influence their attitudes and behaviors regarding healthy eating [15]. Identifying gaps in adolescents’ KAP is essential for implementing effective policies promoting healthy eating behavior.

The eating behavior of individuals is influenced by a multitude of personal, familial, and social factors, which also exhibit distinct regional characteristics [16]. Chongqing, as one of the largest cities in China, stands apart from other cities due to its diverse adolescent population, representing various socioeconomic backgrounds, cultural norms, and dietary habits. The residents of the Chongqing region are recognized for their specific cooking methods, such as stir-frying, frying, and grilling. They have a predilection for spicy and sour flavors, thus often incorporating generous amounts of oil, salt, and chili into their cuisine. Compared to Chongqing, regions such as Jiangsu and Zhejiang, which are also in the south, have lighter tastes [17]. Research indicates that various digestive system disorders, including chronic atrophic gastritis, gastric ulcers, and Helicobacter pylori infection, are associated with unhealthy dietary habits such as heavy oil and salt intake [18]. Among China’s population of 1.4 billion, approximately 120 million individuals suffer from gastrointestinal diseases. Among them, the incidence of gastric ulcers is 10%, and the incidence of chronic gastritis is 30% [19]. Simultaneously, according to the digestive health monitoring data in China, the digestive health score in Chongqing is only 52.22 points (total score of 100) [20]. This indicates that the prognosis for the prevalence of digestive system diseases in the Chongqing region is also not optimistic. Additionally, in the Chongqing area, there is a phenomenon where communal and lively dining is more popular than individual or solitary dining experiences [21]. This unique cultural context of Chongqing further underscores the importance of studying the KAP of healthy eating among adolescents in this region. It aids in identifying culturally sensitive strategies that can effectively promote healthy eating behaviors among Chongqing’s adolescent population.

While some studies have explored the status of KAP related to diet among adolescents in certain regions of China [22,23,24], there remains a dearth of evidence specifically from Chongqing. Moreover, structural equation modeling (SEM) is an analytical technique utilized to untangle intricate relationships and causal pathways while quantitatively assessing the direct and indirect effects of variables, particularly when latent constructs are involved [25,26]. According to the KAP theory, there is a causal relationship between knowledge, attitude, and practices [27,28]. Prior studies have effectively utilized SEM as a valuable tool to explore and understand the current state of KAP in various domains, including nutrition labeling [29], influenza vaccination [30], antibiotic prescribing [31], and dental caries prevention [32]. But studies employing structural equation modeling to investigate the intricate relationships between healthy dietary KAP are exceptionally scarce in this context. Hence, our study aimed to develop a comprehensive structural equation model to analyze the relationships between knowledge, attitudes, and practices regarding healthy eating among adolescents in Chongqing. Furthermore, research has indicated that the educational environment plays a significant role in shaping the nutritional knowledge and attitudes of adolescents [33,34], so we sought to investigate the impact of the educational environment on the KAP of adolescents.

Based on the above research and theoretical foundations, we propose the following five hypotheses (Figure 1):

**Hypothesis** **1 (H1):***Adolescents who have a better educational environment are more likely to achieve higher scores in healthy eating knowledge*.

**Hypothesis** **2 (H2):***Adolescents who have a better educational environment are more likely to exhibit a more positive attitude towards healthy eating*.

**Hypothesis** **3 (H3):***Adolescents with higher scores in healthy eating knowledge are more likely to demonstrate a more positive attitude towards healthy eating*.

**Hypothesis** **4 (H4):***Adolescents who possess a more positive attitude towards healthy eating are more likely to engage in healthy eating practices*.

**Hypothesis** **5 (H5):***Adolescents with higher scores in healthy eating knowledge are more likely to engage in healthy eating practices*.

Through the identification of specific domains where adolescents may lack adequate knowledge, possess negative attitudes, or engage in unhealthy dietary practices, our findings can inform the development of targeted interventions that effectively address these issues.

## 2. Methods

### 2.1. Study Design and Sample Collection

The present cross-sectional study was conducted using the online survey platform “Questionnaire Star” from 2 December to 15 December 2021. The study received support from the Chongqing Municipal Education Commission and employed the convenience sampling method to select 310 secondary schools located in 39 districts and counties of Chongqing as the survey sites. The questionnaire link or QR code was shared with the schools’ WeChat work groups. The designated teachers then forwarded the instructions and questionnaire to parents via the WeChat group for grades 7, 8, and 9. In cases where explicit consent was obtained from both parents and students, with mutual agreement acknowledged through co-signed informed consent, the questionnaire was independently completed by students anonymously at home during weekends or after school.

Questionnaires with outliers and missing data and those with too short a completion time were excluded. In total, 139,832 questionnaires were selected for analysis. Prior to conducting this study, ethical approval was obtained from the esteemed Ethics Committee of Chongqing Medical University, ensuring adherence to rigorous ethical standards (approval number: 2021041). All participants were informed of the study and were given means of consent at the beginning of the questionnaire.

### 2.2. Measures

The questionnaire used in this study was developed on the basis of KAP theory, mainly referring to the Chinese Dietary Guidelines for School-age Children (2016) [35]; the nutrition literacy scale for middle school students in Chongqing, China [36]; the questionnaire of dietary knowledge, attitudes, and practices of residents in Southwest China [37]; and other available data, combined with information on the main problems related to the dietary behavior of middle school students in Chongqing. The questionnaire demonstrated a satisfactory level of reliability and validity, as indicated by an overall Cronbach’s α coefficient of 0.734 and a Kaiser–Meyer–Olkin Measure (KMO) result of 0.770, *p* < 0.01. It comprises four sections encompassing 24 questions, which focus on sociodemographic characteristics, healthy dietary knowledge, healthy dietary attitude, and healthy dietary practices.

The 10 sociodemographic characteristics included in this study comprised age, sex (male/female), ethnicity (Han/other), grade (first grade/second grade/third grade), boarding school (yes or no), residence (urban/rural), only child (yes/no), parents’ education (low: junior high school and below, medium: senior high school or technical secondary school, high: college or bachelor’s degree and above), self-reported height, and self-reported weight. Body Mass Index (BMI) was calculated by dividing the body weight in kilograms by the square of the body height in meters. Additionally, BMI categories were determined using the age- and sex-specific BMI cut-off points recommended by the International Obesity Task Force (IOTF) [38]. In this sociodemographic part, we considered parental educational level and residence as observed variables for the latent variable aspect of the educational environment. A scoring system was employed, where a low parental educational level was assigned 0 points, a medium level received 1 point, and a high level was assigned 2 points. Similarly, for the variable of residence, rural areas were assigned a score of 0 points, while urban areas received a score of 1 point.

Please refer to Appendix A for detailed questions related to KAP. The section on healthy dietary knowledge comprised six questions about the basic nutrition literacy of middle school students in the format of single choice and multiple choice. Each correct choice was assigned 1 point. As the questions in this section were dichotomous variables and varied in their focus, the total score of the knowledge component was treated as an explicit variable, rather than being considered as a latent variable, for inclusion in the path analysis.

Both healthy dietary attitudes and practices included four questions. The options in these two parts are evaluated by a five-point Likert scale, and the options and scores are as follows: strongly disagree/never: 0 points, disagree/occasionally: 1 point, neutral/sometimes: 2 points, agree/often: 3 points, strongly agree/always: 4 points. The higher the score, the more positive the attitude towards a healthy diet and the better the dietary habits. The specific questions were included in each part, and the responses are detailed in the results section.

### 2.3. Statistical Analysis

All statistical analyses and data management were conducted using STATA/MP (version 17.0, College Station, TX, USA). Descriptive statistics were employed to summarize the characteristics of the sample. Categorical variables were described using frequencies and percentages, while the mean (standard error) was utilized for continuous variables. To assess differences, a one-way analysis of variance or chi-square test was employed as appropriate. Pearson’s correlation test was utilized to examine relationships between latent variables. Parameter estimation was carried out using the maximum likelihood estimate (MLE), and non-significant variables identified in the exploratory factor analysis were excluded. Given the large sample size of our study, the chi-square value is a sample-sensitive indicator, and too much post hoc model modification can reduce the chi-square value but make the model distorted. Therefore, the chi-square value was discarded as an indicator of model fit in this study [39,40]. Hence, model fitting was evaluated using the following indices: CFI (comparative fit index), IFI (incremental fit index), GFI (goodness-of-fit index), AGFI (adjusted goodness-of-fit index), NFI (normed fit index), TLI (Tucker–Lewis index), RMSEA (root mean square error of approximation), and SRMR (standardized root mean square residual). The Monte Carlo test, a less time-consuming alternative procedure to bootstrap for testing of the mediated/indirect effect, was employed to assess the significance and calculate the 95% confidence intervals for the mediating effect of relevant variables in the ideal model [41,42]. Statistical significance was determined at *p* < 0.05, indicating a significant difference (two-sided).

## 3. Results

### 3.1. Basic Demographic Characteristics

The basic demographic characteristics of the participants are presented in Table 1. A total of 139,832 participants were included, with 69,788 males and 70,044 females. The average age of participants was 14.8 ± 0.7 years; the vast majority of them were Han Chinese (96.0%); 57.3% were in school accommodation; and 41.1% resided in rural areas. The respondents were mainly distributed across the 7th grade (45.3%) and 8th grade (39.8%). In 76.3% of families, there was only one child. The proportion of father and mother with higher education was relatively low, at 13.8% and 12.6%, respectively. A total of 64.9% participants had a normal BMI. The rate of overweight and obesity was as high as 14%.

### 3.2. Nutrition Knowledge

Table 2 shows the responses to the health and nutrition knowledge section of the survey. The average score for this section was 3.60 ± 1.50, and the correct rate is 60.0%. The highest correct response rate was for dairy product nutrition label recognition, which was 70.7%. The awareness rate of the health risks associated with long-term consumption of sugary beverages was second highest at 70.5%, while the lowest awareness rate, at only 47.5%, was for the knowledge of the six core guidelines of the Chinese Dietary Guidelines for Residents.

### 3.3. Attitudes

Table 3 shows the distribution of the health and nutrition attitudes. The average score was 13.61 ± 2.29. In total, 71.4% of the respondents considered nutrition to be very important for health. Moreover, 55.1% of the respondents were very willing to apply their nutrition knowledge to guide their daily dietary behaviors. In addition, 19.6% of the individuals held a neutral attitude toward the food safety risks associated with street vendors.

### 3.4. Eating Practices

Table 4 shows the frequency distribution of healthy eating practices. The average score was 12.06 ± 3.08. About 54.0 percent of respondents pay attention to the expiration date when buying food, and 31.9 percent of respondents maintain their water intake even when they are not thirsty.

### 3.5. The Test Result of Convergence Validity and Combination Validity

To assess the measurement quality of the latent variables, we report the following indicators as shown in Table 5. The composite reliability (CR) for all latent variables is greater than 0.7, indicating good internal consistency of the latent variables. The standardized factor loadings (Std.) indicate strong associations between the observed variables and the latent variables. Average variance extraction (AVE) is greater than 0.36, indicating that the measured variables have a strong ability to explain the variance of potential variables [43].

### 3.6. The Detection Results of Factor Correlation and Discriminant Validity

The discriminant validity is examined by comparing the square root of the average variance extracted (AVE) with the correlations between constructs. After testing, the square root of AVE for this data is found to be higher than the correlations with other constructs, indicating a good level of discriminant validity, as shown in Table 6.

### 3.7. Model Fitting Indices

In order to evaluate the fit degree of the model, we examined the fit validity indices of the model. The results are shown in Table 7. Through repeated modification and fitting of the model, the fitting indices of SEM (CFI = 0.968, IFI = 0.968, GFI = 0.986, AGFI = 0.978, NFI = 0.968, TLI = 0.958, RMSEA = 0.04, SRMR = 0.029) were better than the corresponding threshold values, indicating that the data have a good degree of fitting with the model.

### 3.8. Structural Equation Modeling Results

Figure 2 shows the final SEM model. Table 8 shows the hypothesis test results of the knowledge, attitude, and practice path coefficients. Table 9 illustrates the significance test of the mediating effect of the final model. The following information can be obtained from Table 2, Table 8 and Table 9. All five hypothesized paths have a significance level of *p* < 0.001, indicating that H1 to H5 can be accepted, specifically, H1 and H2: the educational environment had a significant positive effect on adolescents’ knowledge of healthy eating (β = 0.235, 95% CI: 0.230–0.241) and on attitude (β = 0.143, 95% CI: 0.137–0.150); H3 and H5: knowledge had a significant positive direct effect on attitude (β = 0.225, 95% CI: 0.219–0.231) and on practice (β = 0.054, 95% CI: 0.048–0.060); and H4: attitude had a significant positive direct effect on practice (β = 0.565, 95% CI: 0.559–0.571) (Table 8). The indirect effect of knowledge on practice through attitude was 0.127, 95% CI: 0.124–0.131, *p* < 0.001. The direct effect of knowledge on practice was 0.054, 95% CI: 0.048–0.060, *p* < 0.001. The ratio of indirect effect to direct effect was 2.361, indicating that the indirect effect had a more significant effect on practice than the direct effect (Table 9).

## 4. Discussion

In this study, we developed a comprehensive SEM to analyze the relationship between knowledge, attitudes, and practices regarding healthy eating among adolescents in Chongqing. The study revealed a significant positive relationship between healthy eating-related knowledge and attitudes, attitudes and practices, and knowledge and practices. Our findings align with the fundamental assumption of the KAP theory, which posits a positive relationship among the variables of knowledge, attitude, and practices [44]. Moreover, similar findings have been reported in community trial research conducted in the United States targeting children [45] and cross-sectional studies focused on international students in Ireland [15], as well as cross-sectional studies involving adolescent schoolgirls in Bangladesh [46]. Furthermore, it is noteworthy that although the path coefficient for the direct effect of knowledge on practices is statistically significant, it is relatively small (only 0.054). In contrast, the coefficient for the indirect effect of knowledge on practices with attitude as the mediating variable is 0.127. The magnitude of the indirect effect is significantly greater than that of the direct effect, indicating the crucial role of attitude in the causal chain of knowledge, attitude, and practices [30]. This result means our health knowledge education is effective and suggests that when adolescents receive nutrition education, they are more likely to develop positive beliefs and eventually change their behaviors related to nutrition [47].

Furthermore, our study revealed a significant association between a favorable educational environment and improved knowledge of healthy eating as well as more positive attitudes toward healthy eating among adolescents. These findings align with previous studies [48,49,50]. This suggests that both parental education and the urban environment can play important roles in shaping individuals’ understanding and attitudes regarding healthy eating. In China, most highly educated people live in urban areas [51]. This urban setting provides a favorable environment for the implementation of school-based nutrition education initiatives and targeted public health campaigns aimed at promoting healthy eating habits among adolescents [52]. These interventions focus on delivering information about the importance of balanced nutrition, the benefits of specific food groups, and practical strategies for making healthier choices [53]. Furthermore, parents with higher levels of education in urban areas are often more aware of the significance of healthy eating and actively seek out and utilize available resources and information. They play a crucial role in imparting their knowledge to their adolescent children through conversations, acting as role models, and creating an environment that supports healthy eating habits [54]. Parental education is frequently associated with higher socioeconomic status, which can further influence adolescents’ access to resources and opportunities for healthy eating [55,56]. The advantages of the educational environment in urban areas, as mentioned above, contribute to improved knowledge and positive attitudes toward healthy eating among adolescents. It is imperative to acknowledge that these findings should not undermine the significance of promoting healthy eating behaviors in rural or socioeconomically disadvantaged areas. Access to resources, educational programs, and support for healthy eating should be made accessible and equitable across all communities.

Although the results of this study showed that adolescents’ knowledge, attitudes, and practices of healthy eating were unsatisfactory. However, there are still some results worth noting. Specifically, the correct rate related to nutrition labels was the highest, followed by the awareness rate of diseases caused by long-term consumption of sugary drinks. It may be because teenagers are in a critical period of growth and development, and sugary drinks are popular during their growth. Studies have shown that adolescents are an important target group for nutrition intervention, and the prevalence of obesity and other phenomena has stimulated people’s interest in improving adolescent nutrition education [57]. Therefore, schools and parents have greater publicity and education in this regard, and parents have greater supervision of children’s related eating behaviors. Parents in many countries are encouraging their children to buy prepackaged foods by identifying and understanding nutritional labels [51]. Six countries—Australia, Canada, Chile, Mexico, the United Kingdom, and the United States—have generally higher awareness of nutrition labels among children and adolescents [58]. Compared to other countries, China also needs to strengthen nutrition education on “how to interpret nutrition labels”. For the awareness rate of diseases caused by the consumption of sugary drinks, the prevalence of knowledge about the health status of drinking sugary drinks was similar in different studies. For example, relevant studies in the United States and Australia were similar to the results of this study [59]. However, the discussion on the correlation between knowledge of sugary drinks and the intake behavior related to sugary drinks among adolescents was not carried out in this study. However, this study was not conducted to explore the correlation between knowledge about sugary drinks and the consumption behavior related to sugary drinks among adolescents. The results of this study show that nearly half of middle school students are willing to use knowledge to guide their daily behavior. Future research could further track healthy eating behaviors, such as how often children and adolescents pay attention to nutrition labels when purchasing pre-packaged foods and limiting consumption of sugar-sweetened beverages.

According to the results of this study, more people hold a neutral attitude toward the food safety risks of roadside stalls. In addition to conducting food safety-related education activities for middle school students, the relevant departments should also strengthen the food hygiene and safety awareness of food operators around schools. The owners of “roadside stalls” should be reminded of food safety risks, and food safety knowledge such as fast inspection of edible agricultural products and toxic and harmful products should be propagandized to improve the food safety awareness of “roadside stall” food operators.

The advantages of this study are as follows: firstly, it employs an SEM approach to rigorously examine the theoretical framework rooted in the well-established KAP model. Remarkably, this study stands as the pioneering endeavor utilizing SEM to delve into the intricate interplay between knowledge, attitudes, and practices concerning healthy eating within the adolescent population of Chongqing. Secondly, a notable strength lies in the substantial sample size, ensuring the representativeness and reliability of our research results. However, there are still a few limitations in the present study. Firstly, this study is conducted as a cross-sectional study, which means that it captures data at a single point in time. However, it is important to recognize that dietary behavior change is a complex process that may occur over a long period. Therefore, longitudinal studies are necessary to examine the causal relationship between knowledge, attitudes, and behaviors related to diet. Secondly, it is important to acknowledge that the survey in this study primarily relied on online platforms. However, it is worth considering that relying solely on online platforms may introduce biases and potential inaccuracies in the data collected. Not all individuals have equal access to the internet, and the online sample may not be fully representative of the target population. In addition, this study was about adolescents whose data in terms of height and weight were self-reported, which may lead to the presence of bias due to subjective factors or other reasons. Lastly, it is important to acknowledge that the dimensions of healthy eating KAP are abstract constructs, lacking a widely accepted consensus on precise measurement variables. Consequently, the measurement variables employed in this study may not capture the entirety of knowledge, attitudes, and behaviors regarding a healthy diet. Therefore, further research endeavors should aim to explore and establish a more comprehensive understanding and definition of individuals’ healthy eating knowledge, attitudes, and behaviors.

## 5. Conclusions

The KAP model utilized in this study demonstrates strong applicability for analyzing the healthy eating practices of middle school students in Chongqing. This study highlights that the mediating role of attitude between healthy eating knowledge and practices exerts a more significant indirect positive effect than the direct positive effect of knowledge on practices. This finding underscores the importance of attitude in bridging the gap between knowledge and behavior. Additionally, there is a significant association between a favorable educational environment and improved knowledge of healthy eating as well as more positive attitudes toward healthy eating among adolescents. In the future, it is imperative for nutrition and health education to prioritize the exploration of more effective avenues enabling adolescents to effectively apply knowledge into practical action. Consequently, policies aimed at preventing adolescent obesity should concentrate on implementing impactful nutrition and health education strategies that not only foster positive attitudes but also guide corresponding healthy behaviors.

## Figures and Tables

**Figure 1 nutrients-16-00167-f001:**
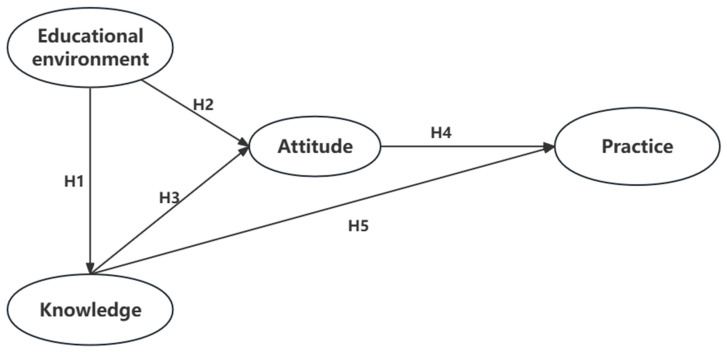
Hypothesis testing of the model.

**Figure 2 nutrients-16-00167-f002:**
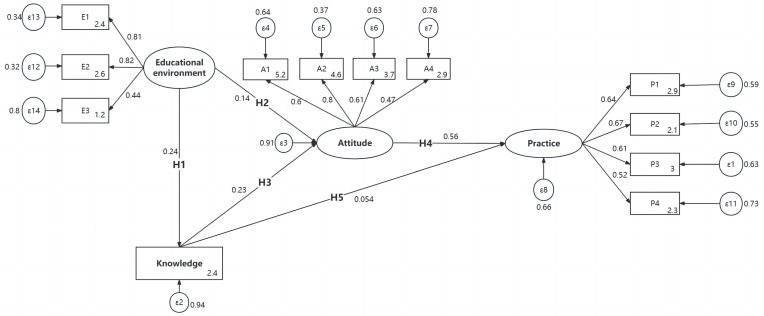
The rectangle represents the measured variable; the ellipse represents the latent variable; the circle represents the residual term; and the value on the arrow of a single item represents the standardization coefficient. All paths were significant (*p* < 0.05). E1, mother’s education level. E2, father’s level of education. E3, family residence. A1, I think it is important to know nutrition for health. A2, I would like to use my nutrition knowledge to guide my daily eating behavior. A3, “Separate dining system” or “public chopsticks and spoons” should be promoted in public restaurants. A4, there are great food safety risks in roadside food stalls. P1, I insist on three meals a day at a fixed time. P2, even when I’m not thirsty, I hydrate myself regularly. P3, when I buy food, I look at the production date and expiration date. P4, when dining in the dining hall, I will inform the staff in advance how much food I want to eat.

**Table 1 nutrients-16-00167-t001:** Basic demographic characteristics of 139,832 participants.

Factor		*n* (%) or Mean (SD)
Age, mean (SD)		14.8 (0.7)
Sex	Male	69,788 (49.9%)
	Female	70,044 (50.1%)
Ethnicity	Han	134,232 (96.0%)
	Others	5600 (4.0%)
Grade	Grade 7	63,312 (45.3%)
	Grade 8	55,695 (39.8%)
	Grade 9	20,825 (14.9%)
Accommodation	Yes	80,136 (57.3%)
	No	59,696 (42.7%)
Residence	Urban	82,369 (58.9%)
	Rural	57,463 (41.1%)
One child	Yes	106,648 (76.3%)
	No	33,184 (23.7%)
Father’s education	Low	92,124 (65.9%)
	Medium	28,443 (20.3%)
	High	19,265 (13.8%)
Mother’s education	Low	95,813 (68.5%)
	Medium	26,337 (18.8%)
	High	17,682 (12.6%)
BMI	Thinness	29,509 (21.1%)
	Normal	90,741 (64.9%)
	Overweight	13,528 (9.7%)
	Obese	6054 (4.3%)

**Table 2 nutrients-16-00167-t002:** Knowledge section score situation.

Items	Wrong	Right	Score, Mean (SD)
K1 ^a^	73,469 (52.5%)	66,363 (47.5%)	0.475 (0.499)
K2 ^b^	40,902 (29.3%)	98,930 (70.7%)	0.707 (0.455)
K3 ^c^	70,073 (50.1%)	69,759 (49.9%)	0.499 (0.500)
K4 ^d^	57,292 (41.0%)	82,540 (59.0%)	0.590 (0.492)
K5 ^e^	41,189 (29.5%)	98,643 (70.5%)	0.705 (0.456)
K6 ^f^	52,707 (37.7%)	87,125 (62.3%)	0.623 (0.485)
Total			3.600 (1.498)

Data are presented as mean (SD) for continuous measures and *n* (%) for categorical measures. ^a^ Six core recommendations mentioned in the Dietary Guidelines for Chinese Residents (2016 Edition). ^b^ Based on the nutritional labels of the following two dairy products, please determine which one belongs to a dairy product with higher nutritional value? ^c^ Which of the statements about daily combinations and consumption of vegetables and fruits is incorrect? ^d^ A salty diet is most likely to cause which diseases? ^e^ Long-term consumption of sugary drinks increases the risk of which diseases? ^f^ What are the advantages of “separate dining system”? (Please refer to Appendix A for detailed questions related to the knowledge section).

**Table 3 nutrients-16-00167-t003:** Distribution of healthy eating attitudes.

Items	Strongly Disagree	Disagree	Neutral	Agree	Strongly Agree	Score, Mean (SD)
A1 ^g^	1238 (0.9%)	558 (0.4%)	4840 (3.5%)	33,415 (23.9%)	99,781 (71.4%)	3.644 (0.655)
A2 ^h^	499 (0.4%)	538 (0.4%)	10,581 (7.6%)	51,179 (36.6%)	77,035 (55.1%)	3.457 (0.682)
A3 ^i^	1565 (1.1%)	2523 (1.8%)	15,100 (10.8%)	41,851 (29.9%)	78,793 (56.3%)	3.386 (0.834)
A4 ^j^	3684 (2.6%)	3662 (2.6%)	27,340 (19.6%)	42,833 (30.6%)	62,313 (44.6%)	3.119 (0.984)
Total						13.606 (2.289)

Data are presented as mean (SD) for continuous measures and *n* (%) for categorical measures. ^g^ I think it is important to know nutrition for health. ^h^ I would like to use my nutrition knowledge to guide my daily eating behavior. ^i^ “Separate dining system” or “public chopsticks and spoons” should be promoted in public restaurants. ^j^ There are great food safety risks in roadside food stalls. (Please refer to Appendix A for detailed questions related to the attitude section).

**Table 4 nutrients-16-00167-t004:** Frequency distribution of healthy eating practices.

Items	Always	Often	Sometimes	Occasionally	Never	Score, Mean (SD)
P1 ^k^	61,427 (43.9%)	46,303 (33.1%)	22,106 (15.8%)	7987 (5.7%)	2009 (1.4%)	3.124 (0.969)
P2 ^l^	44,646 (31.9%)	38,929 (27.8%)	34,629 (24.8%)	17,196 (12.3%)	4432 (3.2%)	2.731 (1.128)
P3 ^m^	75,506 (54.0%)	36,353 (26.0%)	17,728 (12.7%)	8088 (5.8%)	2157 (1.5%)	3.251 (0.987)
P4 ^n^	57,753 (41.3%)	42,801 (30.6%)	20,879 (14.9%)	11,404 (8.2%)	6995 (5.0%)	2.951 (1.156)
Total						12.056 (3.080)

Data are presented as mean (SD) for continuous measures and *n* (%) for categorical measures. ^k^ I insist on three meals a day at a fixed time. ^l^ Even when I’m not thirsty, I hydrate myself regularly. ^m^ When I buy food, I look at the production date and expiration date. ^n^ When dining in the dining hall, I will inform the staff in advance how much food I want to eat. (Please refer to Appendix A for detailed questions related to the practice section.).

**Table 5 nutrients-16-00167-t005:** Measurements of latent variables.

Latent Variables	Items	Unstd.	S.E.	Z-Value	*p*-Value	Std.	SMC	CR	AVE
E	E1	1.000				0.814	0.663	0.746	0.511
	E2	0.982	0.005	189.5	*p* < 0.001	0.822	0.676		
	E3	0.289	0.002	145.53	*p* < 0.001	0.442	0.195		
A	A4	1.000				0.465	0.216	0.715	0.394
	A3	1.103	0.008	140.601	*p* < 0.001	0.606	0.367		
	A2	1.185	0.008	143.745	*p* < 0.001	0.796	0.634		
	A1	0.857	0.006	135.123	*p* < 0.001	0.600	0.360		
P	P1	1.000				0.642	0.412	0.7049	0.3759
	P2	1.212	0.007	181.339	*p* < 0.001	0.669	0.448		
	P3	0.970	0.006	158.184	*p* < 0.001	0.612	0.375		
	P4	0.963	0.007	142.287	*p* < 0.001	0.519	0.269		

E, educational environment; A, healthy eating attitudes; P, healthy eating practices; Unstd., unstandardized; Std., standardized; S.E., standard error; SMC, squared multiple correlation; CR, composite reliability; AVE, average variance extracted.

**Table 6 nutrients-16-00167-t006:** Factor correlations and discriminant validity.

Latent Variables	E	A	P
E	**0.715**		
A	0.193	**0.628**	
P	0.133	0.576	**0.613**

E, educational environment; A, healthy eating attitudes; P, healthy eating practices. The bold value represents the square root of AVEs.

**Table 7 nutrients-16-00167-t007:** The fit indices of the structural equation model (SEM).

Fit Index	The Goodness-of-Fit Index of SEM							
	CFI	IFI	GFI	AGFI	NFI	TLI	RMSEA	SRMR
Ideal standards	>0.90	>0.90	>0.90	>0.90	>0.90	>0.90	<0.08	<0.08
Measurement value	0.968	0.968	0.986	0.978	0.968	0.958	0.040	0.029

CFI, comparative fit index. IFI, incremental fit index; GFI, goodness-of-fit index; AGFI, adjusted goodness-of-fit index; NFI, normed fit index; TLI, Tucker–Lewis index; RMSEA, root mean square error of approximation; SRMR, standardized root mean square residual.

**Table 8 nutrients-16-00167-t008:** Hypothesis testing results for path coefficients of knowledge, attitude, practices, and educational environment.

Hypothesized Paths	Unstd.	S.E.	Z-Value	*p*-Value	Std.	95% CI		Results
						LLCI	ULCI	
H1: K←E	0.470	0.006	79.000	*p* < 0.001	0.235	0.230	0.241	Accepted
H2: A←E	0.087	0.002	40.386	*p* < 0.001	0.143	0.137	0.150	Accepted
H3: A←K	0.069	0.001	66.642	*p* < 0.001	0.225	0.219	0.231	Accepted
H4: P←A	0.767	0.007	111.198	*p* < 0.001	0.565	0.559	0.571	Accepted
H5: P←K	0.022	0.001	17.582	*p* < 0.001	0.054	0.048	0.060	Accepted

Unstd., unstandardized; S.E., standard error; Std., standardized; 95% CI, 95% confidence interval; LLCI, lower limit of confidence interval; ULCI, upper limit of confidence interval.

**Table 9 nutrients-16-00167-t009:** Direct and indirect effect (via attitude) of knowledge on practice.

Parameter	Estimate	95% CI		*p*-Value
		LLCI	ULCI	
Indirect effect	0.127	0.124	0.131	<0.001
Direct effect	0.054	0.048	0.060	<0.001
Indirect effect/Direct effect	2.361			

95% CI, 95% confidence interval; LLCI, lower limit of confidence interval; ULCI, upper limit of confidence interval.

## Data Availability

The datasets generated and/or analyzed during the current study are not publicly available due to funding requirements but are available from the corresponding author on reasonable request.

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
