# Peer review of "Understanding the Knowledge, Attitudes, and Practices of Healthy Eating among Adolescents in Chongqing, China: An Empirical Study Utilizing Structural Equation Modeling"

_nutrients, 2024, doi:10.3390/nu16010167_

Round 1

Reviewer 1 Report

Comments and Suggestions for Authors

I was pleased to read the manuscript entitled "Understanding the Knowledge, Attitudes, and Practices (KAP) of Healthy Eating Among Adolescents in Chongqing, China: An Empirical Study Utilizing Structural Equation Modeling(SEM)" and to review it.

The study developed a comprehensive structural equation model (SEM) to analyze the relationship between knowledge, attitudes, practices (KAP), and healthy eating among adolescents in Chongqing (China). It revealed a significant positive relationship between healthy eating-related knowledge and attitudes, attitudes and practices, as well as knowledge and practices. The results of the study can be applied in planning healthy eating strategies.

From a scientific point of view, the article did not reveal particularly new regularities, but it was interesting to read it because the article describes the SEM modelling procedure in detail and reasonably, which is relatively new in research in the field of KAP. From this point of view, the article can be used as a teaching tool for students delving into SEM methods, especially the analysis of mediating factors.

The article is written in a typical format. Only a few comments or suggestions can be made.

Title – the title is informative and it accurately reflect the manuscript, however abbrevations are recommended not to be included in title.

Abstract – the abstract is more or less complete and adequately reflects the content of the manuscript. However, it is not good that the findings replicate the results. This needs to be corrected in order for the findings to generalize the results.

Introduction – the Introduction provide sufficient theoretical background for the study. All examined research questions and/or hypotheses were introduced and backed by literature. Relevant and unbiased literature was used. The introduction is structured logically and the text is fluent. The rationale of the study is well described and the study problem is stated clearly.

Method – all important aspects of the methods (sampling, ethics, measures, analyses) are clearly described and are appropriate to answer the proposed research questions.

Results – in general, results are clearly organized and presented.

- Table 1. There is no need to compare males and females, so the table can be narrowed down.

- Table 8. Column title and note. Probably 'CR' must be changed to 'Z-value'.

Discussion – the structure of the Discussion is clear. The interpretations is appropriate and is supported by the results. The study findings are discussed with relevant literature and within the limits of the study. However, the Discussion forgets to discuss whether the hypotheses H1 to H5 were confirmed.

Conclusions – are supported by the results.

References - please pay more attention to the requirements of referring for MDPI publishers (e.g. abbreviation of journal names).

Thank you for considering my opinion. I encourage authors to keep on working to improve the manuscript.

Reviewer 2 Report

Comments and Suggestions for Authors

Τhank you for the opportunity to review this paper

This is an interesting study entitled: Understanding the Knowledge, Attitudes, and Practices (KAP) of Healthy Eating Among Adolescents in Chongqing, China: An Empirical Study Utilizing Structural Equation Modeling (SEM)

An interesting study with an innovative and relevant methodology is described.

There are some topics in the article that should be considered:

Regarding the methodology

The bioethical issues should be described in more detail  mentioning about the consent  signed by the parents, not by the children who participated in the study.

Regarding the results

In relation to the results in Table 1, how is it possible that the age variable has a statistical difference between participants of different genders when the median value  is the same in all groups?

Please explain to the readers what is “separate dinning system” (table 2 and 3)

Please describe the content of question A related to knowledge in nutrition (table 2)

The legend of figure two should be more explicit, including the questions, abbreviations and the meaning of the numbers mentioned.

How confounding factors was managed during statistical analysis, for example  the percentage of the body mass index was taken into account as confounding factor for its effect?

 Concerning the discussion

 lIn the limitations of the study, it should be mentioned that declaring weight and height by the adolescent students may be wrong, causing bias to this variant.

 .

Reviewer 3 Report

Comments and Suggestions for Authors

Using a unique sample of Chinese adolescents from Chongqing (People’s Republic of China, PRC), this paper explores the influence of multiple factors (KABP: Knowledge, Attitude, Belief, and Practices) on adolescent healthy eating habits. A web-based questionnaire was administered to 139,832 middle school students in Chongqing from December 2 to December 15, 2021. SEM models were employed to gauge the relationships among the educational environment, knowledge of healthy eating, attitudes, and practices. The sociodemographic characteristics included in this study comprised age, sex (male/female), ethnicity (Han/other), grade (first grade/second grade/third grade), boarding school (yes or no), residence (urban/rural), only child (yes/no), parents' education (low: junior high school and below, medium: senior high school or technical secondary school, high: college or bachelor's degree and above), self-reported height, and self-reported weight. The bootstrap method was utilized to evaluate the significance of mediating effects. The mediating role of attitude between knowledge of healthy eating and practices exerted a more substantial indirect positive impact than the direct positive effect of knowledge on practices. A meaningful connection was established between a favorable educational environment and enhanced knowledge, as well as positive attitudes toward healthy eating among adolescents. Moving forward, the authors conclude that nutrition and health education should prioritize devising effective strategies to translate knowledge into tangible practices.

It is hard not to agree with the authors that promoting healthy eating is vital for adolescent development. However, the manuscript fails to explain the methodology of the study. First, it is not clear why this study is guided by KABP framework. This framework is practically unknown outside of PRC. Second, the choice of bootstrap method is not fully explained. You state chi-square test tends to be biased in large-sample studies. Please provide a reference for this statement. As far as I know, low sample sizes may lead to biased results. Chi-square test is as good as any other post-hoc test employed in this study (GFI, etc.). You also mention (lines 72-75) that “the residents of the Chongqing region are recognized for their specific cooking methods, such as stir-frying, frying, and grilling. They have a predilection for spicy and sour flavors, often incorporating generous amounts of oil and salt into their cuisine.” But is it not true about all southern Chinese cuisine? As far as I know, Chongqing is one of the ‘furnaces of China’. The climate of this city (one of the largest in PRC) is very hot and humid in China. There is also a case of urban pollution which adds to high morbidity from respiratory disease. As for digestive morbidity, I do not process recent data. If you have recent statistics showing high morbidity from digestive disorders, I strongly suggest mentioning it in your paper. This will strongly support your case.  

Round 2

Reviewer 3 Report

Comments and Suggestions for Authors

You have tried to answer my concerns to the best of your abilities.